# Test re-test reliability and construct validity of the star-track test of manual dexterity

Niels Kildebro[1], Ilda Amirian[1], Ismail Gögenur[2] and Jacob Rosenberg[1]

[1] Center for Perioperative Optimization, Department of Surgery, University of Copenhagen, Herlev Hospital, Herlev Ringvej, Herlev, Denmark
[2] Køge Hospital, Department of Surgery, Lykkebækvej, Køge, Denmark

## ABSTRACT

**Objectives.** We wished to determine test re-test reliability and construct validity of the star-track test of manual dexterity.

**Design.** Test re-test reliability was examined in a controlled study. Construct validity was tested in a blinded randomized crossover study.

**Setting.** The study was performed at a university hospital in Denmark.

**Participants.** A total of 11 subjects for test re-test and 20 subjects for the construct validity study were included. All subjects were healthy volunteers.

**Intervention.** The test re-test trial had two measurements with 2 days pause in between. The interventions in the construct validity study included baseline measurement, intervention 1: fatigue, intervention 2: stress, and intervention 3: fatigue and stress. There was a 2 day pause between each intervention.

**Main outcome measure.** An integrated measure of completion time and number of errors was used.

**Results.** All participants completed the study (test re-test $n = 11$; construct validity $n = 20$). The test re-testshowed a strong Pearson product-moment correlation ($r = 0.90$, $n = 11$, $P < 0.01$) with no sign of learning effect. The 20 subjects in the construct validity trial were randomized to the order of the four interventions, so that all subjects completed each intervention once. A repeated measures ANOVA determined that mean integrated measure differed between interventions ($p = 0.002$). Post hoc tests using Bonferroni correction revealed that compared with baseline all interventions had significantly higher integrated scores ranging from 47–59% difference in mean.

**Conclusion.** The star track test of manual dexterity had a strong test re-test reliability, and was able to discriminate between a subject's normal manual dexterity and dexterity after exposure to fatigue and/or stress.

Corresponding author
Niels Kildebro,
nielskildebro@gmail.com

## BACKGROUND

A surgeon's manual dexterity is often an outcome parameter in studies examining environmental effects such as work environment or night shifts on surgeons

(*Dorion & Darveau, 2013*; *Amirian et al., 2014*). Simulation tools are often used, but these are mostly time consuming tests that are not readily available. Often a study needs a tool that is easy to administer and is portable so that it can be used where the study calls for it. One such device was used in an interventional study for measuring surgeons' accuracy (*Dorion & Darveau, 2013*). The surgeons were to follow a star-shaped track with a pair of surgical scissors and each time the scissors touched the border of the track, an error was counted. The track was to be completed 3 times and errors were noted. The study stated that it was examined for test re-test reliability. The authors reported a Pearson's correlation of $r = 0.955$ (*Dorion & Darveau, 2013*; *Savoie & Prince, 2002*).

However, the test needs further validation if it is to be used in further research (*Fess, 1995*). The method of examining the test re-test reliability of the test has not been described previously. The psychometric qualities have not been tested thoroughly enough to state that the test is valid and measures the intended characteristics (*Fess, 1995*; *Law, 1987*; *Rudman & Hannah, 1998*). Furthermore, the test has no equipment construction standards or instructions of use available, so the test lacks repeatability and reproducibility (*Fess, 1995*; *Law, 1987*; *Rudman & Hannah, 1998*). It is necessary to further explore the reliability and validity of the test. With these parameters established, the test could be an excellent tool for measuring manual dexterity, and may be useful in future studies, providing an assessment tool that requires short time to be administered and is commercially available.

The purpose of this article was to provide construction standards, instructions for application, test-retest reliability and construct validity for the star-shaped test of manual dexterity.

## METHOD

### Equipment construction standards

The star-track test of manual dexterity consists of the following components:

- Replacement star Model 32532A from Lafayette Instruments (Lafayette instruments, Loughborough, Leicestershire, UK)
- MakeyMakey from joylabz.com (*JoyLabz, 2014*).
- Computer running the software Star Track 32bit.exe (*Nørregaard, 2014*).
- Standard Metzenbaum surgical scissors

The replacement star Model 32532A from Lafayette Instruments is a metal plate measuring 22 cm × 22 cm with a star-shaped track in its center. The six-pointed star-shaped track measures 15.3 cm from point to opposite point. The track is 0.9 cm wide and is made of a non-conducting material. A MakeyMakey is an inventor kit that can turn everyday objects into touchpads and combine them with a computer. This is explained further at http://makeymakey.com. The MakeyMakey is used to connect the metal plate and the Metzenbaum surgical scissors to a computer. In Fig. 1A, detailed measurements of the dimensions of the metal star-shaped track are shown. A complete setup of the test with all its components is illustrated in Fig. 1B.

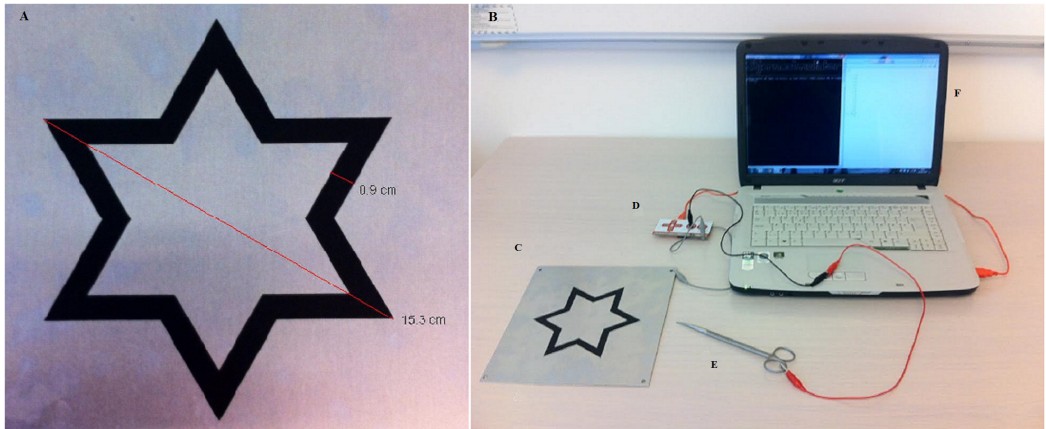

**Figure 1 The test setup.** Picture (A) shows detailed measurements and dimensions of the metal plate with the star-shaped track. Picture B shows a complete setup of the test with all its components. (C) Shows the star-shaped track used in the test. (D) Shows the The MakeyMakey. (E) Shows the standard Metzenbaum surgical scissors and (F) shows the computer running the software Star Track 32bit.exe. Photographs by Niels Kildebro.

## Instructions for administration of the test

The test is set up in the following way. The program Star Track 32bit.exe needs to be installed on the computer used for the test. It is recommended that a laptop be used to increase the transportability of the test. Star Track 32bit.exe is freeware that can be downloaded and installed from http://bitbucket.org/lassebn/star-track. MakeyMakey is used to connect the components of the star-track test using the following steps: 1: Connect the MakeyMakey to the computer via a USB cable. The MakeyMakey will auto install (*JoyLabz, 2014*). 2: Connect the "space" part of the MakeyMakey to the Metzenbaum surgical scissors using two alligator clips linked together. 3: Connect the "ground" of the MakeyMakey to the side of the metal plate using an alligator clip.

The test should be performed in a quiet room without distractions. The examiner places the metal plate about 10 cm from the edge of a table where the subject is sitting comfortably. The examiner instructs the subject to use the scissors in the hand that he wishes to examine. Using the scissors, the subject must follow the star-shaped track ten times, five times clockwise and five times counterclockwise. All ten rounds are completed continuously. The tip of the scissors must be in contact with the star-shaped track during the entire test. Each time the scissors come into contact with the border of the track, an error is registered. Completion time and number of errors are registered automatically by Star Track 32bit. The examiner should read the following instructions to the subject: "To complete the test, you must follow the star-shaped track with the surgical scissors. You are to complete ten rounds, five rounds clockwise and five rounds counterclockwise. All ten rounds are to be completed continuously. You are to complete the ten rounds as quickly as possible with as few errors as possible. An error is counted every time the scissors touch the border of the star-shaped track. The scissors must touch the plate at all times during the test."

With Star Track 32bit running on the computer, the examiner names the test result file. When he/she presses the enter button, the test will begin. When the subject completes the final round, the examiner presses "q" to stop the test.

### Scoring

The Star Track 32bit program automatically records the time (in seconds) it takes to complete the test. It also automatically records errors.

## CONSTRUCT VALIDITY

### Design

We wished to study whether the star-track test would be able to distinguish between the base level of manual dexterity of a person and the level of manual dexterity when the person was fatigued and/or stressed. This was done by conducting a randomized crossover study. Each subject was to complete the star-track test four times. At each trial they were randomly assigned to different interventions. Each subject was to complete all four interventions, and never the same intervention more than once. Each trial was separated by two days' pause. The interventions were: Baseline measurement, the subject completed the star-track test without further intervention. Intervention 1: The subject was fatigued in his dominant arm before completing the star-track test. Intervention 2: the subject was stressed while performing the star-track test. Intervention 3: The subject was fatigued prior to the star-track test and stressed while performing the star-track test. Completion time and number of errors in the star-track test were measured at all four interventions using the Star Track 32bit software. The order in which each subject received the four interventions was randomized (http://www.randomization.com). Using the randomization list, a research fellow, who was not involved in the study, packed and sealed 4 opaque envelopes (labeled day 1, day 4, day 7 and day 10) containing the randomization labels for each subject. These envelopes were opened on the respective days just before the test commenced, so that the subject and examiner were blinded until that point.

### Subjects for construct validity

We aimed to include 20 subjects. The subjects were all volunteers and gave written informed consent before inclusion. They were recruited from local universities, dormitories and hospitals by advertising with posters. Subjects had to understand Danish (written and spoken). They were excluded if they were diagnosed with heart, endocrine, neurological, autoimmune or psychological disease, suffered from sleep disorders or had muscular-skeletal disorders of the upper extremities (e.g., osteoatrosis, rotator cuff syndrome, hand injuries).

### Method of achieving muscular fatigue

The fatigue was achieved by letting the subject hold a 2.5 kg weight in the dominant hand, and holding the dominant arm to 90° flexion. They were to hold this position without moving for as long as possible. The subjects then proceeded to complete the star-track test within 10 s. This test has previously been used to measure muscular fatigue

(*Dorion & Darveau, 2013*) and is described in occupational health literature as a way to achieve static muscular fatigue (*Chaffin, 1973*).

## Method of inducing stress

The brain can focus on performing a specific task at normal level, as long as the mental resources exceed the demands of the task in progress. If multiple tasks are to be performed at the same time, the demands of the tasks will at some point exceed the mental workload tolerance. This will cause stress and subjects will begin making errors (*Boles & Law, 1998*; *Grier et al., 2008*). According to the theory of multiple resources, there are several mental resource pools, enabling several actions to be performed simultaneously. However, if the actions performed require resources from the same pool, they will cause stress more quickly (*Wickens, 2008*). This allows for prediction of workload overload by determining difficulty of the tasks undertaken and task interference. The star-track test is visually perceived, requires spatial understanding and a manual response. The distraction was designed to drain from these mental resources.

While the subject was performing the star-track test, the examiner would show the subject 10 cards from a regular deck of cards. One card per round completed in the star-track test. The card was placed near the metal plate of the star-track test, allowing the subject to have both the star-track and the card in his field of vision. The subject had to identify the card by rank and suit while performing the star-track test. According to the computational 3-D + 1 model of multiple resources (*Wickens, 2008*), the difficulty of the tasks are both simple (following the star-track and identifying cards). The tasks share demands of workloads at two levels (perception and cognition). This gives a total interference of 4 (on a scale of interference from 0–8) (*Wickens, 2008*). If the star-track test is able to detect this workload overload, a higher integrated score (longer completion time and/or more errors), compared with baseline should be scored while completing the test with distraction.

## Test re-test reliability

The reliability was tested with a controlled design. The purpose of this test was to determine the test-retest effect and whether or not the test was consistent over time. The subjects completed the star-track test with an interval of two days between tests. The test conditions were the same on each day of the trial. This design has been used previously to perform test-retest trials of manual dexterity (*Aaron & Jansen, 2003*). Completion time and errors were measured at both tests using the Star Track 32bit software. To measure face-validity, each subject was asked if he understood the purpose of the star-track test, and what they believed it was supposed to measure. We aimed to include 11 subjects. The method of recruitment, as well as inclusion and exclusion criteria were the same as for the validity test.

## Ethics and permissions

The study was registered at Clinicaltrials.gov (NCT02146443). The data collection was approved by the Danish Data Protection Agency (journal no: HEH-2014-060, I-Suite

no. 02972). The study was exempt from approval by The Regional Danish Committee on Biomedical Research Ethics (protocol no: H-6-2014-031). All subjects volunteered and gave written informed consent prior to their participation. No compensation was offered for participating in the study.

## Statistics

All statistics were calculated using IBM SPSS Statistics version 22 (SPSS, Chicago, Illinois, USA) and Microsoft Office Excel 2007. To receive a complete estimate of a subject's manual dexterity, we used an integrated measure for completion time and number of errors (*Silverman, O'Connor & Brull, 1993*). The total number of ranks for conducted trials were found (80 for validity; 22 for the test-retest) and mean rank was calculated. The difference of completion time and number of errors from respective mean ranks was calculated as a percentage in difference, and added on a per-subject basis to form an integrated measure (*Silverman, O'Connor & Brull, 1993*). Since this was a pilot test, no sample size was calculated as no data were available. Thus, sample size was determined by means of qualified estimate (*Hertzog, 2008*). Study population age was described as median (range). We used the Shapiro–Wilk test of normality to determine that data were normally distributed. Mauchly's Test was used to test for Sphericity. We used repeated measures ANOVA with post hoc testing with Bonferoni correction for intergroup measurements in the validation study. The Bonferroni corrected *p*-value was calculated multiplying the least significant differences by the total number of tests possible. Repeated measures ANOVA was selected because the dependent variable (integrated measure of error and time) was measured on the same group of people using different independent variables (interventions). The Pearson correlation coefficient was used for test re-test reliability analysis. Test days were also compared with paired samples *t*-tests. $P \leq 0.05$ was regarded as statistically significant.

## RESULTS

### Construct validity

A total of 20 subjects completed this study, 9 females and 11 males, with a median age of 26 years (range 22–29). Of the participants 3 were left-handed and 17 were right-handed. The subjects were students ($n = 15$), doctors ($n = 2$), nurses ($n = 1$) and an engineer ($n = 1$). The integrated measures scores for each of the four test arms of the crossover study can be found in Table 1. We tested for normality using the Shapiro–Wilk test. It showed that the data for all four test arms of the crossover study did not violate the assumption of normality (baseline $p = 0.73$; intervention 1 $p = 0.67$; intervention 2 $p = 0.79$; intervention 3 $p = 0.44$). A repeated measures ANOVA was done to determine if the integrated measures significantly differed from each other. Mauchly's Test of Sphericity indicated that the assumption of sphericity of the data had not been violated ($\chi 2\ (5) = 7.01$, $p = 0.22$) and thus no correction was used in the repeated measures ANOVA. It was determined that mean integrated measure differed significantly between interventions ($p = 0.002$). Post hoc tests using Bonferroni correction revealed that compared with baseline all interventions

**Table 1 Mean integrated measures of validity test.** Integrated measure of time and error during completion of the star-track test of manual dexterity during each of the four test arms.

| Test arm | N | Integrated measure | SD |
|---|---|---|---|
| Baseline | 20 | −0.39 | 0.51 |
| Intervention 1 | 20 | 0.08 | 0.61 |
| Intervention 2 | 20 | 0.10 | 0.39 |
| Intervention 3 | 20 | 0.20 | 0.55 |

**Table 2 Post hoc tests of repeated measures ANOVA.** Integrated measure of time and error during completion of the star-track test of manual dexterity. Baseline compared to the three interventions.

| Comparison | Mean difference | p-values |
|---|---|---|
| Baseline-intervention 1 | −0.47 (−0.93; −0.02) | 0.04 |
| Baseline-intervention 2 | −0.49 (−0.92; −0.07) | 0.02 |
| Baseline-Intervention 3 | −0.59 (−1.09; −0.09) | 0.01 |

**Notes.**
Values are presented as mean difference in integrated measure with 95% confidence interval. *p*-values calculated with post hoc tests using the Bonferroni correction.

**Table 3 Mean integrated measures of test re-test trial.** Integrated measure of time and error during completion of the star-track test of manual dexterity.

| Test re-test day | N | Mean integrated measure | SD |
|---|---|---|---|
| Test day 1 | 11 | 0.07 | 0.62 |
| Test day 2 | 11 | −0.05 | 0.72 |

had significantly higher integrated scores, indicating that the test was able to differentiate between the baseline and the interventions (see Table 2). Furthermore, intervention 3 scored higher integrated measure than intervention 1 and 2, with a mean difference in integrated measure of 0.12 and 0.10 respectively (see Table 1), although this difference was statistically insignificant ($P = 1$ for both).

## Test re-test reliability

A total of 11 subjects completed this study: 6 males and 5 females. The median age was 27 years (range 22–35). Two of the subjects were left-handed and nine were right-handed. The subjects included 7 students, 1 engineer and 3 nurses. The integrated measures scores for each test day are presented in Table 3. A Pearson product-moment correlation was run to determine the relationship between the test days. The data showed no violation of normality (Shapiro–Wilk test of test day 1 $p = 0.32$ and test day 4 $p = 0.25$), linearity or homoscedasticity. There was a strong, positive correlation between the integrated measures of the two test days ($r = 0.90, n = 11, P < 0.01$).

Test day 1 and test day 4 were compared with paired samples $t$-tests to ensure that the Pearson correlation coefficient was not high due to a consistent difference (e.g., learning effect). There was no significant difference in integrated measure ($p = 0.21$).

## DISCUSSION

The star-track test was able to detect a difference between the baseline measurement and all three of the interventions in the construct validity study. This indicates that the test was able to discriminate between a person's baseline and impaired manual dexterity due to fatigue and/or stress. The test re-test reliability showed that the star-track test had a strong test re-test reliability.

The purpose of the star-track test was to be an evaluative tool for measuring manual dexterity, and to measure changes in individuals. More specifically, the target population of the test was subjects with no impairment or disease in the upper extremities. The test involved a surgical instrument, it was meant to be used in future research to evaluate the manual dexterity of surgeons. If the test is to be used as a descriptive tool, more studies should be conducted where normative data should be collected on different groups of subjects standardized for age, gender, surgical experience and maybe various impairments of the upper extremities. The star-track test has no predictive value yet. To gain this, studies where surgeons' integrated measures in the star-track test are compared to patient outcomes would be needed.

The reliability of the star-track test of manual dexterity has been explored in previous studies (*Dorion & Darveau, 2013*; *Savoie & Prince, 2002*). They reported a Pearsons correlation with $r = 0.955$. In this article, we confirmed the previous findings of a strong test-retest reliability of the star track test, as our finding of $r = 0.90$ was consistent with the previous research. Also, the trials ruled out any significant learning effects, which is especially important for the consistency of the test. Furthermore, we described the method of obtaining the reliability results in detail, which had not been done before. To make the star-track test accessible, equipment construction standards were provided along with instructions for administration. We used components for construction of the test that were commercially available, so that the test can be reconstructed and reproduced. This further established the consistency and reliability of the test. As the data of the test were gained by means of a computer program, we did not examine for inter-rater and intra-rater reliability, as the standardized computer program minimized these factors.

All subjects easily understood that the test measured manual dexterity, which indicated that the test had good face validity and was easy to understand.

The content validity had already been established, as the test was used to test the accuracy of surgeons in a previous study (*Dorion & Darveau, 2013*). However, manual dexterity is a more complete measure of a surgeon's skill than accuracy. Dexterity is the ability to manipulate objects with your hands with a specific purpose in mind (*Dunn et al., 1994*; *Baum & Edwards, 1995*). Dexterity can be subdivided into a static phase, and a dynamic phase which involves powergrip (adapting hand strength) and precision handling of handheld objects (*Kamakura et al., 1980*). The characteristics of manual dexterity are

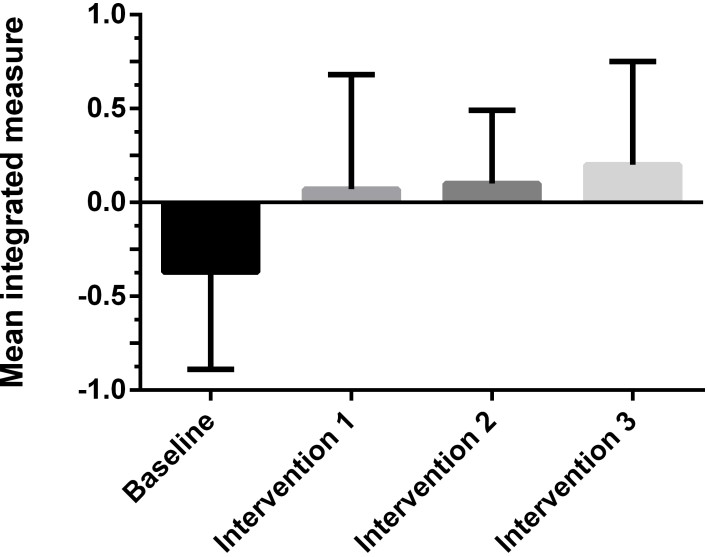

**Figure 2 The mean integrated measures of time and error during completion of the star-track test in the construct validation study.** Integrated measure is percent from mean integrated measure of study population. Whiskers represent standard deviation. A positive score is a poorer than average performance (e.g., longer completion time and/or more errors) when compared to the mean score, while a negative score is better than average.

accuracy and speed (*Aaron & Jansen, 2003*). All this is needed in instrument handling and therefore important to surgeons' technical skills (*Memon et al., 2010*). In this study we expanded the measurement to be a more complete concept of manual dexterity by using an integrated measure of errors and completion time. We believe that by doing this we have increased the content validity of the star-track test.

Construct validity was explored in this study. The trend of the data showed that the more stressed and/or fatigued a subject was, the higher the integrated measure of manual dexterity. This trend is illustrated in Fig. 2. It detected a mean difference of 10%–12% between intervention 3 and both intervention 1 and intervention 2. This indicated that the test might be able to measure different intensities of stress and the effect of fatigue on the subjects' manual dexterity. However, the difference was not statistically significant, and therefore we were unable to confirm this trend. Further testing with a similar setup and a larger sample size should be conducted to further investigate this.

The criterion validity of the test still needs to be established. This could be done in future studies comparing data from the star-track test to other established accuracy tests and tests of manual dexterity.

With the data presented in this article, we believe that the star-track test of manual dexterity may be used in future research when testing the accuracy and manual dexterity of surgeons. The star-track test can be used to discriminate between a subject's normal manual dexterity and after exposure to fatigue and/or stress.

### Funding

The study was partly funded by the Danish Working Environment Authority. The funders had no role in study design, data collection and analysis, decision to publish, or preparation of the manuscript.

### Grant Disclosures

The following grant information was disclosed by the authors:
Danish Working Environment Authority.

### Competing Interests

The authors declare there are no competing interests.

### Author Contributions

- Niels Kildebro conceived and designed the experiments, performed the experiments, analyzed the data, wrote the paper, prepared figures and/or tables, reviewed drafts of the paper.
- Ilda Amirian, Ismail Gögenur and Jacob Rosenberg conceived and designed the experiments, reviewed drafts of the paper.

### Human Ethics

The following information was supplied relating to ethical approvals (i.e., approving body and any reference numbers):

The data collection was approved by the Danish Data Protection Agency (journal no: HEH-2014-060, I-Suite no. 02972).

The study was exempt from approval by The Regional Danish Committee on Biomedical Research Ethics (protocol no: H-6-2014-031).

### Supplemental Information

Supplemental information for this article can be found online at http://dx.doi.org/10.7717/peerj.917#supplemental-information.

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
