# Peer review of "Test re-test reliability and construct validity of the star-track test of manual dexterity"

_PeerJ, doi:10.7717/peerj.917_

## Round 0.1 · original submission · Major Revisions

· Academic Editor

Major Revisions

Please address all the comments and resubmit.

Reviewer 1 ·

Basic reporting

The article meets all the basic requirements.

Experimental design

The designs used in this paper are clearly described and solidly implemented, which are the main strength of this paper.

Validity of the findings

The main concern for this paper is the relatively smaller sample size and the subsequent conclusions. Specific suggestions regarding this concern can be found in the attachment.

Additional comments

Overall, this paper is solid. It serves very well as a starting point for further research. However, the authors need to clarify some of the statistical results, and adjust their conclusions accordingly.

Annotated reviews are not available for download in order to protect the identity of reviewers who chose to remain anonymous.

Reviewer 2 ·

Basic reporting

No Comments

Experimental design

No Comments

Validity of the findings

No Comments

Additional comments

In this manuscript “Test re-test reliability and construct validity of the star-track test of manual dexterity”, the authors attempted to employ star-track test to detect a difference between the baseline measurement and all three of the interventions in the construct validity study. The experimental design is relatively reasonable, the results and discussion are plausible. Furthermore,
The manuscript is easily readable, concise and to the point piece of writing. The language is also acceptable. In addition, there still exist some problems below for further instruction and modification in the manuscript:
1. The Background needs some improvement. In Background, the authors have stated a lot of research achievements obtained by previous researchers, but it seems to be lack of the importance describing on the necessity of the study. Besides, the logicality of the background section is poor, it should be rearranged.
2. The reason for selecting repeated measures ANOVA (line167, page9) but not One-Way ANOVA or two-way ANOVA is not explained.
3. Why are authors declaring “Furthermore; intervention 3 scored higher integrated measure than intervention 1 and 2, with a mean difference in integrated measure of 0.12 and 0.10 respectively (see figure 1), although this difference was statistically insignificant (P = 1 for both)” in line 185~188, page 10? This difference isn't statistically significant, could you explain it?
4. Are the results of test-retest trials about manual dexterity in the paper consistent with other scholarly research? The authors should consider conducting comparative analysis with previous studies by others.

Reviewer 3 ·

Basic reporting

Redundant figures (for example pictures 1a, 1d and 1e). There should be labels for picture 1c. Grammatical (inappropriate use of past and present tenses) and spelling errors. Reporting literatures in inappropriate sections for example method for inducing stress.

Experimental design

Surgery can be really complex and I do not see how a star-track test would be useful for further studies. Authors wrongly quote the reference by Dorion & Darveau 2013. Why choose a scissor and not others? There is no mention who were the experimental subjects, were they students and how they were sampled. Not clear what was being done for test-retest – interventions not repeated? Also, two days may be too short for test-retest.

Validity of the findings

Sample population is relatively young, should cover more age ranges since there can be young and older surgeons.

Additional comments

I agree with the importance to test for dexterity of surgeons but surgery is not as straight-forward as a star-track test. Besides dexterity, experience does count. Furthermore, small sample sizes and age ranges are not representative. Authors will need to check their grammars.

---

## Round 0.2 · accepted · Accept

· Academic Editor

Accept

Thanks for the thorough revision, which resulted in a much stronger manuscript. It is now accepted for publication.

Reviewer 1 ·

Basic reporting

The paper is solid in basic reporting.

Experimental design

The design is sound.

Validity of the findings

The findings are valid based on the correct experimental design and the appropriate statistical tools used in this paper.